# Associations between Parental Educational Attainment, Children’s 24-h Behaviors and Children’s Hyperactivity Behavior in the COVID-19 Pandemic

**DOI:** 10.3390/healthcare12050516

**Published:** 2024-02-21

**Authors:** Meiyuan Chen, Michael Chia, Terence Chua, Zhi Shen, Mengke Kang, Lu Chen, Tiantian Tong, Xiaozan Wang

**Affiliations:** 1College of Physical Education & Health, East China Normal University, Shanghai 200241, China; meiyuanchen@stu.ecnu.edu.cn (M.C.);; 2Physical Education & Sports Science Academic Group, National Institute of Education, Nanyang Technological University, Singapore 637616, Singapore; 3Department of Astronomy, Shanghai Jiao Tong University, Shanghai 200240, China; 4School of Physical Education, Shandong University, Jinan 250061, China; 5College of Sports, China University of Mining and Technology, Xuzhou 221000, China; tongtt@cumt.edu.cn

**Keywords:** Children’s 24-h behaviors, Children’s hyperactivity behavior, Parental educational attainment

## Abstract

Background: Parental Educational Attainment and children’s 24-h behaviors significantly influenced children’s hyperactivity symptoms. This study aimed to examine the mediating role of children’s 24-h behavior changes due to the COVID-19 pandemic between Parental Educational Attainment and children’s hyperactivity index. It also aimed to investigate the associations between Children’s Physical Activity, digital media use, sleep, and hyperactivity index between two clusters of Parental Educational Attainments. The goal was to provide targeted behavioral optimization recommendations for caregivers to reduce the risk of children’s hyperactivity. Methods: The study was a collaborative extension of the International iPreschooler Surveillance Study Among Asians and otheRs project and the Chinese Children and Adolescent Sports Health Promotion Action Project. The Parent-Surveillance of Digital Media in Childhood Questionnaire^®^ and the Abbreviated Rating Scales from the Conners Parent Symptom Questionnaire were used to measure Parental Educational Attainment, children’s behavior changes during the COVID-19 pandemic, and hyperactivity indexes. A total of 11,190 parents of 6-to-12-year-old children completed the online surveys in mainland China. A structural equation model was established by using Smart-PLS, and the linear regression model, and isotemporal substitution models were established by using a Compositional Data Analysis package with R program to achieve the research objectives. Results: Changes in children’s 24-h behaviors due to the COVID-19 pandemic had a significant mediation effect on the negative associations between Parental Educational Attainment and children’s hyperactivity index (β = 0.018, T = 4.521, *p* < 0.001) with a total effect (β = −0.046, T = 4.521, *p* < 0.001) and a direct effect (β = −0.064, T = 6.330, *p* < 0.001). Children’s Digital Media use was significantly and negatively associated with hyperactivity index among all children. Reallocated time from digital media use to both sleep and physical activity decreased the hyperactivity index, and vice-versa. For parents without tertiary education (R^2^ = 0.09, *p* < 0.001), sleep was significantly and negatively associated with the hyperactivity index (β_ilr-CSL_ = −0.06, *p* < 0.001); for parents with tertiary education (R^2^ = 0.07, *p* < 0.001), physical activity was significantly and negatively associated with the hyperactivity index (β_ilr-CPA_ = −0.05, *p* < 0.001), and sleep was significantly and positively associated with the hyperactivity index (β_ilr-CSL_ = 0.03, *p* < 0.001). A significant increase in the hyperactivity index was detected when physical activity time was reallocated to sleep, with a significant decrease in the opposite direction. Conclusions: Parental Educational Attainment and children’s 24-h behaviors directly influenced children’s hyperactivity index. However, a purposeful and targeted optimization of children’s 24-h behaviors—namely, physical activity, digital media use, and sleep—could assist parents with different educational attainments to reduce their children’s hyperactivity index and mitigate the risk of hyperactivity.

## 1. Introduction

Hyperactivity refers to increased movement, such as excessive fidgeting, taps, or talking at inappropriate times [1,2], and the Children’s Hyperactivity Index (CHI) represents the degree of hyperactive behavior and the risk of hyperactivity that a child might have [3]. Hyperactivity is not only an important diagnostic criterion of Attention Deficit Hyperactivity Disorder (ADHD) [4,5], but it also reflects some degree of mental or physical and behavioral disorders [1,2]. Hyperactive behavior usually appears in childhood and continues into adulthood without intervention. Such behaviors in childhood and adolescence affect social interaction, academic performance, and mental health, posing a serious challenge to the development of families, communities, and countries [6,7]. Therefore, it is important to address and provide timely attention and intervention for hyperactive behaviors in children.

### 1.1. Social-Ecological Theory and Children’s Hyperactive Behavior

The multi-level context in which behavioral change and healthy development occurs must be taken into account [8]. Social-ecological theory suggests that individuals and their environments interact and that individual behaviors are influenced not only by the macrosystem (such as national public policy), but also directly by the microsystem (including the roles played by parents and teachers) [9]. This concept is highly significant for addressing and preventing public health problems, and it finds wide application in health promotion [8,10]. Similarly, hyperactive behavior in children is influenced not only by intrinsic complex factors such as genes and neurodevelopment, but also by external factors such as family and society [2]. However, it has become increasingly challenging to change internal factors, making external factors an important starting point for helping children improve their hyperactive behavior.

The social-ecological theory identifies the family as the most influential and the most proximal system, with parents being an important direct factor influencing children’s health as their primary guardians [11]. The social-ecological theory also indicates that the child-parent relationship affects a person’s behavior [12]. Therefore, we should pay attention to the role of parents in improving children’s hyperactive behavior. Additionally, the social-ecological theory emphasizes public policy, which is closely related to the social environment, and policies based on the overall environment invisibly influence children’s behavior [8,10]. The prevalence of the Coronavirus Disease 2019 (COVID-19) in recent years has impacted not only the macro-policies in various countries and regions around the world, but also educational initiatives for children at different levels such as schools and communities. Understanding children’s hyperactivity during the COVID-19 pandemic along with other daily behaviors, is beneficial with regard to proposing strategies to improve children’s hyperactivity. The social-ecological theory should be applied by focusing on at least two levels of influence [13]. The present study will explore health promotion strategies for children’s hyperactive behaviors by addressing two factors: parents and the COVID-19 pandemic.

### 1.2. Parental Educational Attainment and Children’s Hyperactive Behavior

In terms of microsystems of the social-ecological theory, children’s healthy behaviors are strongly and directly influenced by parents. Some research shows that home environment and parental factors are associated with hyperactivity behavior [14,15,16,17,18], such as parental emotions were significantly correlated with children’s hyperactivity [19]. However, it is known that parents’ behavior is greatly affected by Parental Educational Attainment (PEA), which is an important predictor of children’s behavior, educational outcomes, and other aspects [20,21].

PEA was defined as the highest level of formal education attended by children’s parents, either the mother or father or both [22,23]. A significant negative association between PEA and symptoms or positive rates of ADHD in children has been reported [24,25,26]. It appears that higher levels of PEA are linked to lower rates of behavioral problems and increased positivity in children with ADHD. Hyperactivity behavior is also a main clinical characteristic of ADHD. Thus, it is evident that PEA influences the parenting style that parents adopt and may have far-reaching effects on children’s hyperactive behaviors.

Therefore, it is valuable to examine how individual factors like children’s behaviors, combined with PEA, are associated with hyperactive behavior. This examination can lead to evidence-based recommendations for behavior improvement tailored to specific groups of children.

### 1.3. Children’s Hyperactivity and Behavior Changes during COVID-19 Pandemic

Within the macrosystems of the social-ecological model, the health and behaviors of children all over the world have been profoundly affected by the COVID-19 pandemic and related policies. Emergent research shows that children’s ADHD symptoms were exacerbated during the COVID-19 pandemic [27,28], with noticeable increases in attentiveness and hyperactive behavior [29], and children’s 24-h behavior patterns were also altered. Children’s 24-h behavior typically refers to the various behavioral manifestations of individuals over 24-h, including various types of physical activity, sedentary behavior, sleep, etc. It is widely acknowledged that factors like lack of sleep, reduced physical activity, excessive sedentary behavior, and screen time are strongly linked to behavioral issues in children [30,31,32,33,34].

Moreover, Children’s Physical Activity (CPA) saw a significant decrease due to increased digital media use (CDM), and sleep duration (CSL) markedly increased during the COVID-19 pandemic. These shifts led to exacerbated problems in both typically developing children and those with ADHD [27,35]. Similarly, hyperactive behavior was affected, although there are fewer studies on this aspect during the COVID-19 pandemic [36,37,38,39,40,41]. The Centers for Disease Control and Prevention in the USA recommends that children with ADHD should manage their symptoms and maintain health by engaging in physical activity, limiting screen time, and ensuring sufficient sleep [42]. These strategies are also effective not only in managing hyperactive behavior, but also in promoting overall well-being [32,33,43,44,45,46].

It has been established in prior studies that there is a close association between children’s hyperactive behavior, Parental Educational Attainment, and children’s 24-h behavior. However, it is notable that, although there have been many studies discussing the relationship between children’s hyperactive behavior and 24-h behavior with parents’ education attainment, there are apparently no studies interrelating children’s daily behaviors with the two and analyzing their intrinsic associations and influencing pathways. Additionally, CPA, CSL, and CDM behavior in research are routinely treated as independent variables; however, these daily behaviors are interdependent over the course of the 24-h day [47]. Therefore, it is important to consider significant behaviors engaged throughout each 24-h cycle as a whole in explaining children’s performance or health condition. Compositional Data Analysis (CoDA) is a well-established statistical method that takes into account all components of a whole simultaneously by employing a set of logs while investigating the association between different temporal components and health outcomes through modeling [47,48,49], and is a superior approach to delve deeper into the intrinsic relationship between behavior and health outcomes.

Based on the above research status, this study proposed two hypotheses and comprehensively verified them through model construction. Primarily, the associations between parental education, 24-h behavioral changes of children during the COVID-19 pandemic, and child hyperactivity index were explored to elucidate the importance of parental education and optimizing children’s behaviors in reducing the risk of hyperactivity. Secondarily, the dose-dependent relationships between 24-h behavioral patterns and the risk of child hyperactivity at different PEA was analyzed by using CoDA, with a full consideration of the 24-h behavioral patterns. Targeted optimization recommendations were provided for parents with different educational backgrounds to reduce the risk of hyperactivity and to promote general health and well-being in children. The hypotheses are as follows:

**H1:** 
*Behavioral changes in children’s 24-h behaviors during the COVID-19 pandemic are mediated by PEA and CHI. PEA will influence children’s 24-h behavioral changes during the COVID-19 pandemic and further influence CHI.*


**H2:** 
*Targeted optimization of 24-h behaviors can effectively reduce children’s hyperactivity behavior for parents with different educational attainment.*


## 2. Methods

### 2.1. Procedures and Participants

This study was a collaborative extension of the International iPreschooler Surveillance Study Among Asians and otheRs (IISSAAR) (iissaar.com, accessed on 15 November 2023) project and the Chinese Children and Adolescent Sports Health Promotion Action Project (CCASHPAP) [50]. The IISSAAR research was an international initiative examining the behaviors of preschool children in collaboration with researchers from countries such as Japan, Indonesia, South Korea, Finland, and other countries. The IISSAAR questionnaire was adapted and validity- and reliability-tested for use on primary school children in the present study. Additionally, CCASHPAP was a comprehensive program designed to promote healthy child development through a range of physical activity-based interventions. The research conducted for this study received approval from the Committee on Human Research Protection of East China Normal University, China (HR 405-202) and was carried out in September 2022.

The research utilized a random sampling method, and the questionnaire link was distributed to schools to parents of 6–12-year-old children across different regions of mainland China as a component of the “Chinese Children and Youth Sports and Health Promotion Action Project” through an online survey. The schoolteachers then forwarded the questionnaire link to the parents of the primary school pupils, who were given the option to participate at their discretion. Ultimately, 13,588 parents participated in the survey and, after data cleaning, the dataset comprised 11,190 valid responses (82.3%) (Appendix A [51,52]).

### 2.2. Survey Instruments

#### 2.2.1. Parent-Surveillance of Digital Media in Childhood Questionnaire (Parent-smalQ^®^)

Parent-smalQ^®^ (Appendix A) is an adaption from a questionnaire known as the Surveillance of digital Media hAbits in earLy chiLdhood Questionnaire (SMALLQ^®^), which was initially developed by Chia and his colleagues [53]. SMALLQ^®^ targeted parents of children in middle to late childhood. The Parent-smalQ^®^, unlike a psychometric instrument, serves as a lifestyle questionnaire that gathers information from parent respondents about their children’s behavioral habits related to screen media use, physical activity, and sleep. Structured into three sections, the Parent-smalQ^®^ covers (i) digital media use and parents’ perceptions of their child’s digital use; (ii) habits involving the child’s physical play and sleep habits; and (iii) general details about the parent and child. Instances of child digital media use on weekdays and weekends encompass activities like accessing information, communication, content creation, entertainment, and school-related work. Parents were queried about their children’s physical play habits on weekdays and weekends, along with general information about the child (e.g., gender, age, body mass index-for-age classification) and the parent (e.g., child relationship, highest educational attainment). The Parent-smalQ^®^ underwent a forward- and back-translated from its original language (English) to simplified Mandarin for implementation in China. The simplified Mandarin Parent-smalQ^®^ underwent reliability and validity assessments, resulting in satisfactory overall reliability (Cronbach’s α = 0.887) and validity (χ^2^/df = 3.28, Root Mean Square Error of Approximation = 0.047). Additional data and details can be found in Appendix A.

#### 2.2.2. Abbreviated Rating Scales from the Conners Parent Symptom Questionnaire (PSQ)

The Conners Scale has undergone multiple revisions since its original development, with the 1978 revision being the most extensively utilized version in China [54]. Comprising 48 items, it encompasses 5 subscales for conduct problems, learning problems, psychosomatic problems, hyperactivity/impulsivity, and anxiety. Additionally, the questionnaire’s 10 questions form a concise symptom assessment that individually evaluates the Children’s Hyperactivity Index (CHI). This aspect allows for the utilization of the questionnaire in diverse epidemiological investigations [55]. Its international utilization is widespread [56,57].

### 2.3. Data Processing

#### 2.3.1. Behavior Changes in Children’s Physical Activity, Sleep, and Digital Media (CPA, CSL, and CDM) Because of the COVID-19 Pandemic

Parents were asked, “How has the average amount of time your child spent each day on physical activity/sleep/digital media changed from the time immediately before COVID-19 pandemic restrictions were imposed to the time now?” (at the time of questionnaire completion) within the Parent-smalQ^®^. The associated questionnaire items regarding CSL, CPA, and CDM times had Likert-type response choices, spanning from 1 (significant decrease) to 5 (significant increase). Notably, due to the anti-epidemic policies in China, the question was further supplemented. During the specialist validity testing, all experts unanimously concurred that the question was aligned with the prevailing situation in China and warranted inclusion.

#### 2.3.2. Children’s Physical Activity, Sleep, and Digital Media (CPA, CSL, and CDM)

Parents were asked to provide specific time durations for each behavior reported. For instance, they were prompted with questions such as, “Think about the average amount of time your child spent playing outside of school in the past seven days?”; “Think about how long your child usually slept during naps and night each day for the past seven days?”; “Think about the average amount of time your child spent on digital media screens outside of school in the past seven days?” These questions encompassed both weekday and weekend responses, in addition to delineating various activities (e.g., recreation, study, etc.) that transpired within a 24-h timeframe. The final duration for each behavior was determined by averaging the responses from both weekdays and weekends. To assess the reliability of the data, reliability tests were conducted (Cronbach α_CPA_ = 0.73; Cronbach α_CSL_ = 0.77; Cronbach α_CDM_ = 0.73). With all Cronbach alpha values greater than 0.7, the data are considered to be reliable and can be analyzed [58].

#### 2.3.3. Parental Educational Attainment (PEA)

Parental Educational Attainment (PEA) was determined using the question, “What is your highest level of education?” This question presented respondents with various choices, including “no formal education”, “elementary school”, “secondary school”, “post-secondary”, “bachelor”, and “graduate”. Each option was assigned a numerical value from 1 to 6, with higher values corresponding to higher levels of education attainment. Subsequently, parents were categorized into two distinct clusters based on their PEA: parents with tertiary education (PTE) and parents without tertiary education (PWTE). Within this classification, PTE encompassed answer choices 4 to 6, while PWTE encompassed answer choices 1 to 3.

#### 2.3.4. Children’s Hyperactivity Index (CHI)

Children’s Hyperactivity Index (CHI) was assessed and evaluated using the 10-item abbreviated symptom questionnaire in Chinese (Appendix A) [54]. The scale employs a four-point scoring system ranging from 0 to 3. For the calculation of the Z score, the item scores are summed and then divided by the total number of items. On this scale, a score of 0 signifies no problem, 1 indicates sporadic or mild issues, 2 reflects frequent or severe challenges, and 3 indicates very frequent or severe issues. A higher CHI score corresponds to an increased risk of hyperactive behavior in a child. The threshold value for the hyperactivity index is 1.5, and a score of ≥1.5 suggests the potential presence of hyperactivity [54].

### 2.4. Data Analysis

The reliability and validity tests for the Parent-smalQ^®^ were initially conducted utilizing reliability and factor analyses, and were executed with IBM SPSS Statistics (Version 26.0, IBM Corp, Armonk, NY, USA) and AMOS software (Version 23.0, IBM Corp, Armonk, NY, USA). Additionally, the characteristics and differences among the children with varying levels of PEA in terms of age, gender, geography, and behaviors are presented in Table 1. To address potential common method bias, Harman’s single-factor test was employed. The variation explained by the primary component was determined to be 30%, which falls below the threshold of 40%. This finding indicates the absence of significant common technique bias within the data.

Subsequently, in order to explore the potential mediation effect of behavior changes in CPA, CSL, and CDM due to the COVID-19 pandemic on the association between PEA and CHI, a structural equation model applied the partial least squares (PLS) was conducted, the resultant model assessed both the direct effect and total effect between PEA and CHI, as well as the indirect effect arising from behavior changes in children due to the COVID-19 pandemic. All analyses were carried out using the Smart-PLS (Version 3.2.9) as in other studies [59,60,61]. In terms of model structure and metrics, the factor loadings, Cronbach’s alpha, average variance extracted (AVE), and scale composite reliability (SCR) were used to assess the reliability and convergent validity of the structure [62], and the Fornell–Larcker ratio and the Heterotrait–Monotrait ratio of correlations to assess the discriminant validity of the constructs [62]. Multicollinearity among variables (VIF) was also calculated for all indicators to assess multicollinearity among variables [62]. For model fitting, the approximate fit metrics like the Standardized Root Mean Square Residual (SRMR) value, the Normed Fit Index (NFI), and the Root Mean Square Residual covariance (RMS-theta) were evaluated [62]. Finally, calculate the main effect path in the structural modeling coefficient and its significance with the “bootstrap” option with 5000 replications, and A 95% confidence interval was established to assess the significance of the effects [62].

To provide intentional advice on optimizing children’s behaviors for different PEA groups, linear regression models and isotemporal substitution models were generated by applying Compositional Data Analysis (CoDA) for children with PTE and PWTE, and the predicted-change graphs were plotted. The CoDA is a well-established statistical method [63], that explores the association between various behaviors and health outcomes by considering people’s behavioral time as a whole and the relative changes in the time spent on different behaviors in epidemiology [48,49,64,65]. In the present study, the CPA, CSL, and CDM time of children were considered as a whole, and the CHI was the health outcome variable. The ratio log of time use components serves as the foundation of the CoDA application [48], and CPA, CSL, and CDM were further modeled using isometric log-ratio (ilr) data transformation. The specific methodological process and principles can be found in the brief guide by Chastin et al. [49]. The linear regression assumptions were checked for linearity and normality of distributions, isotemporal substitution models were retained with 95% confidence intervals, and all model results were adjusted for gender, age, and geography. All analyses were completed through the R program (Version 4.2) and the open-source packages for computing we use are compositions and robCompositions. The level of significance was accepted at *p* < 0.05.

## 3. Results

### 3.1. Sample Description and Descriptive Analyses

The sample characteristics and variables are outlined in Table 1. The study encompassed a total sample size of 11,190 participants. Among these, 341 individuals (3%) exhibited hyperactive tendencies, while the remaining 10,876 (97%) displayed non-hyperactive tendencies. The participants in the present study came from 7 primary regions in mainland China, including 13 provinces, as detailed in Figure 1. In addition, 2211 participants were fathers (19.7%), of which 584 were PWTE and 1627 were PTE; 8767 fillers were mothers (78.3%), of which 2418 were PWTE and 6049 were PTE; and 212 fillers were guardians (2%), of which 104 were PWTE and 108 were PTE.

**Table 1 healthcare-12-00516-t001:** Sample characteristics and descriptive statistics for variables (N (%) or M (SD)).

Characteristics		Total (N = 11,190)	Hyperactivity Tendency (N = 314)	Non-Hyperactivity Tendency (N = 10,876)
Age		8.65 (1.85)	8.44 (1.71)	8.66 (1.85)
Sex	girl	5795 (41.30)	199 (63.40)	5596 (51.50)
	boy	5395 (38.50)	115 (36.6)	5280 (48.50)
Geography	Northeast China	1819 (16.26)	49 (15.61)	1770 (16.27
	North China	1362 (12.17)	37 (11.78)	1325 (12.18)
	East China	2094 (18.71)	38 (12.10)	2056 (18.90)
	Northwest China	160 (1.43)	7 (2.23)	155 (1.40)
	Southwest China	3629 (32.43)	133 (42.36)	3494 (32.13)
	South China	1912 (19.00)	42 (13.38)	1869 (17.18)
	Central China	214 (1.91)	8 (2.55)	207 (1.90)
Parental Educational Attainment	No formal education	49 (3.00)	3 (1.00)	46 (0.40)
	elementary school	404 (2.90)	22 (7.00)	382 (3.50)
	secondary school	2953 (21.1)	96 (30.60)	2857 (26.30)
	post-secondary	4405 (31.4)	116 (36.90)	4289 (39.40)
	bachelor	2759 (19.7)	62 (19.70)	2697 (24.80)
	graduate	620 (4.4)	15 (4.80)	605 (5.60)
**Independent variable**		**Total** **(N = 11,190)**	**PTE** **(N = 3406)**	**PWTE** **(N = 7784)**
The change because of the COVID-19 pandemic	C19-CPA	2.79 (0.94)	2.77 (0.93)	2.85 (0.95)
	C19-CSL	3.11 (0.68)	3.09 (0.64)	3.14 (0.72)
	C19-CDM	3.52 (1.01)	3.57 (0.98)	3.4 (1.05)
Behavior Time	CPA	126.04 (71.32)	126.69 (71.56)	124.57 (70.76)
	CSL	569.61 (51.98)	569.17 (50.69)	570.61 (54.53)
	CDM	59.50 (39.50)	57.17 (39.89)	64.82 (40.38)
Hyperactivity Index		0.46 (0.43)	0.46 (0.42)	0.50 (0.45)

C19 = a change because of the COVID-19 pandemic; CPA = Children’s Physical Activity; CSL = Children’s Sleep; CDM = Children’s Digital Media use; PTE = Parent with Tertiary Education; PWTE = Parent Without Tertiary Education.

### 3.2. Structural Equation Model Result

#### 3.2.1. Assessment of the Measurement Model and the Structural Equation Model

The results of construct reliability and validity are presented in Table 2. The factor loadings of CHI are all above 0.60 for each item and are significant. The Scale Construct Reliability (SCR) of CHI was 0.987, surpassing the threshold of 0.7, and the Average Variance Extracted (AVE) was 0.501, which is above the recommended level of 0.5. These findings align with established criteria, signifying strong scale reliability and validity [66]. Furthermore, discriminant validity was assessed using the Fornell–Larcker ratio and Heterotrait–Monotrait ratio of correlations (HTMT). Table 2 illustrates that the inter-correlations between variables remained below the square root of AVE, and all values of HTMT are below 0.85, indicating satisfactory discriminant validity [66,67].

Furthermore, the Variance Inflation Factor (VIF) values were calculated for each variable and presented in Table 3. Notably, all VIF values were below 3, indicating an absence of multicollinearity in the data [66]. It is notable that in the model, variables like PEA, C19-CPA, C19-CSL, and C19-CDM are linked to only one observed variable, and their SCR values, AVE values, factor loadings, and Cronbach’s alpha values are all displayed as 1, so not presented.

The model fit results are presented in Table 4. The SRMR value is 0.050, which is less than the standard value of 0.08. The NFI value is 0.899, which is higher than the standard of 0.8 and very close to 1. The RMS-Theta is 0.119, which is less than the standard of 0.12. In addition, the d_ULS is 0.260, the d_G is 0.075 and the Chi-square is 5101.864. The values indicate that the model has a reasonable degree of adaptation [67,68].

#### 3.2.2. The Effected Path in the Structural Equation Model

The results of the structural equation modeling are presented in Figure 2 and Table 5, and these results validate Hypothesis H1. For the entire model, there was a significant negative correlation between PEA and CHI in the total effect (β = −0.046, T = 4.521, *p* < 0.001), as well as a significant negative correlation between PEA and CHI in the direct effect (β = −0.064, T = 6.330, *p* < 0.001), and the changes in child behavior resulting from the COVID-19 pandemic significantly mediated this process (β = 0.018, T = 9.063, *p* < 0.001), demonstrating that parents with lower educational attainment were effectively able to reduce CHI by optimizing their children’s CPA, CSL, and CDM. In terms of mediation pathways, there were a total of seven, all of which exhibited significant mediation effects.

### 3.3. The Line Regression Model and the Isotemporal Substitution Model Results

The regression analysis revealed specific associations within different PEA groups. For PWTE (R^2^ = 0.09, *p* < 0.001), a negative and significant association was observed between CHI and CSL (β_ilr-CSL_ = −0.06, *p* < 0.001). For PTE (R^2^ = 0.07, *p* < 0.001), CHI showed a negative and significant association with CPA (β_ilr-CPA_ = −0.05, *p* < 0.001), while a positive and significant association with CSL (β_ilr-CSL_ = 0.03, *p* < 0.001). Notably, in both models, CDM exhibited a negative association with CHI (β_PTE-CDM_ = −0.04, *p* < 0.001; β_PWTE-CDM_ = −0.05, *p* < 0.001).

Figure 3 and Table 6 present the findings of the isotemporal substitution model, depicting trends and outcomes. For PWTE, reallocation of CSL time to CDM in 15-min increments led to a significant increase in CHI by 0.62%, 1.24%, 1.86%, and 2.27%, respectively (percentages are derived by dividing the CHI change values in Table 3 by the baseline, the same as below). When CDM time was reallocated to CSL or CPA, CHI significantly decreased by 0.82%, 1.86%, 3.09%, 4.95%, and 0.62%, 1.24%, 2.27%, and 3.92%. However, the reallocation of CPA time to CDM and CSL showed trends of increase and decrease in CHI for PTE, but these changes were not statistically significant. In the case of PTE, reallocation of CPA and CSL time to CDM in 15-min increments led to significant increases in CHI by 1.56%, 2.90%, 4.45%, 5.79%, 0.89%, 1.56%, 2.23%, and 2.67%. Conversely, these reallocations resulted in significant decreases in CHI by 1.78%, 3.79%, 6.46%, 11.14% and 1.11%, 2.44%, 4.68%, 8.91%. When CPA time was reallocated to CSL, CHI significantly increased by 0.67%, 1.34%, 2.23%, and 3.12%, and significantly decreased by 0.67%, 1.34%, 1.78%, and 2.23%, and vice versa.

## 4. Discussion

The present study further extends the previous research by effectively validating the previous hypotheses through model construction and analysis. In addition to reaffirming the important mediating roles of CPA, CSL, and CDM behavior change in PEA and CHI based on children’s behavior changes during COVID-19, it is notable that the relationship between CPA, CSL, CDM, and CHI and the corresponding improvement was also found to vary across PEA groups, which, in turn, required different targeted improvements.

### 4.1. PEA, CHI, and Children’s Behavioral Changes Because of the COVID-19 Pandemic Are Deeply Intertwined

The research findings demonstrated a significant and negative correlation between PEA and CHI, indicating that higher PEA levels are associated with lower CHI scores. These results align with other related studies [24,26,69,70]. This observation could plausibly be attributed to parents with higher educational attainment possessing better cognitive and behavioral management strategies for their children, which subsequently influence and mitigate hyperactive behaviors. Existing studies highlighted the pivotal role of positive parenting as a protective factor for children with ADHD [71,72] and by deduction hyperactivity. Notably, each individual behavior (CPA, CSL, and CDM) displayed an independent association with CHI. However, a substantial total indirect effect stemming from behavior changes in CPA, CSL, and CDM due to the COVID-19 pandemic was also observed. This implies that children’s behavior is influenced by PEA, and improvements in CHI can be achieved as parents exert an influence over and enhance their children’s behavior. Changes in these behaviors played a partial mediation role, reflecting the intricate interplay between PEA, behavior changes, and CHI. This is one of the important new findings of the present study. The global impact of the COVID-19 pandemic has led to varying degrees of behavior changes among children. Within the social-ecological model, both the microsystem and the macrosystem are pivotal factors affecting children’s health [9,10]. In this context, parents must remain vigilant and proactive in managing their children’s health and behaviors to prevent an escalation in CHI levels. Parents and caregivers need to be directly or indirectly supported in these endeavors, with context-specific solutions that are sustainable.

### 4.2. Associations of CPA and CSL with CHI Varied across PEAs and Necessitate Targeted Intervention

In the mediation effect model, alterations in both CPA and CSL due to the COVID-19 pandemic exhibited significant negative correlations with CHI, while changes in CDM were significantly and positively correlated with CHI (see Table 1 and Figure 2). These findings align with prior research results [27,73,74]. However, what is intriguing is that when we computed the time allocation for children’s behaviors and constructed a linear regression model based on PEA clusters, the associations between children’s behaviors and CHI displayed variations. Among children from PWTE, CSL time exhibited a significant negative association with CHI, while CPA did not. Conversely, among children from PTE, CPA showed a significant negative association with CHI, while CSL exhibited a significant positive association with CHI. These findings are both novel and unprecedented. Previous research primarily examined the association between the overall PEA, children’s behavior, and CHI. In contrast, this study demonstrated that the relationship between CSL and CHI differs across PEA clusters, suggesting potential variations in CSL based on individual health statuses.

Within the context of China, it is conceivable that PTE parents may adopt more effective child-rearing practices due to their superior knowledge. However, PTE parents might also subject their children to heightened performance expectations in various domains, potentially compromising healthy behaviors. PA is recognized for its efficacy in regulating physical and mental well-being [43,75,76]. In the demanding environment of intensive learning, children might require increased PA to manage psychological stress and emotions, ultimately aiding in reducing CHI [77]. Intriguingly, while traditionally accepted views propose a negative association between adequate sleep and hyperactivity levels in children, our research results unveiled a notable positive correlation between CSL and CHI among children with PTE. In contrast, the relationship between CSL and CHI among children with PWTE was consistent with established views. An in-depth analysis of the data reveals that CSL among children with PTE averages about 9.5 h, well within the guidelines of 7–12 h set by the American Academy of Sleep Medicine [78]. However, excessive or insufficient sleep and sleep quality also play crucial roles in overall health and well-being [79]. Although children with PTE adhere to the sleep duration guidelines, potential issues with sleep quality might persist. Nonetheless, this does not necessarily imply a need for a substantial reduction in CSL for children with PTE. The isotemporal substitution model demonstrated that for children with PTE, CHI significantly decreased when CSL time was reallocated to CPA, but CHI increased significantly when CSL time was allocated to CDM. Likewise, for children with PWTE, reallocating CSL time to CDM also resulted in a significant increase in CHI. Our research underscores the importance of CSL for both PTE and PWTE children, with a higher importance of CPA for PTE children. Still, further research is warranted to comprehensively understand the intricate relationship between CSL and CHI. The factors influencing children’s hyperactive behavior are multifaceted and diverse. The social-ecological model highlights the direct influence of parents, corroborated by our study’s findings that the impact of behavioral changes varies across children with different PEA backgrounds. As a result, tailored and effective recommendations should be offered to assist parents in directly intervening in their children’s behavior to mitigate hyperactivity risk.

### 4.3. Associations between CDM and CHI Are Consistent across PEA and Parents and Caregivers Need to Get More Support to Monitor CDM

It is noteworthy that the current study’s findings revealed a significant and negative association between CDM time and CHI among children belonging to both PTE and PWTE clusters. This observation validated and reinforced the mediation effect of CDM changes within the mediation model, which demonstrated the most substantial impact. Modern digital media, with its attributes of instant feedback, immediate gratification, and high-speed stimuli, particularly in activities like cyber-gaming, are believed to intensify hyper-reactivity and hyperactivity tendencies in children, whether or not they have ADHD [80]. Engagement in cyber-gaming, watching videos, or other screen-related behaviors can expose vulnerable children to the risk of gaming addiction, posing a serious threat to their overall development and well-being. Experts emphasize that excessive and untimely exposure to screens can negatively impact children’s health and behavior [81,82]. Notably, the reported CDM time in the study significantly surpassed the World Health Organization’s recommendation of no more than 60 min of daily screen time. This time frame pertains to extracurricular and non-school screen engagement. Before the COVID-19 outbreak, Chinese students typically attended cultural courses indoors for about 35–45 min per class, including a 10 min break between classes, and participated in other interest programs such as sports after school, and teachers rarely used electronic screens throughout the day. However, after COVID-19, students often participated in all of their classes throughout the day with the help of computers and other electronic devices online. It is evident that the changes brought about by the COVID-19 pandemic, including alterations in national education and schooling policies, have led to a notable surge in online lessons and virtual schooling. Consequently, the prevalence of CDM time is likely to continue escalating. This, in turn, elevates the risk of hyperactivity in children. Thus, the vigilant supervision of children’s CDM activities by parents and caregivers becomes increasingly paramount [83], especially considering the influence of the macro-environment on individual behaviors as proposed by the social-ecological theory. In this regard, parents, teachers, and caregivers must receive sustainable and context-specific support to mitigate the insidious threats to children’s health and wellbeing for years to come.

### 4.4. Strengths and Limitations

This study had several notable strengths. Primarily, it pioneered investigating the mediation effect between PEA and CHI, focusing on behavioral changes induced by the COVID-19 pandemic. Another strength of the research is the utilization of CoDA and the isotemporal substitution model to scrutinize the dose-dependent relationship between children’s behaviors and CHI. In particular, this research pioneered the examination of the associations between CPA, CSL, CDM, and CHI times from a compositional analysis standpoint.

However, certain limitations warrant acknowledgment. The research relied on subjective child-proxy or parent-reported data concerning CPA, CSL, and CDM times, which could be susceptible to recall bias and social desirability biases. To mitigate these limitations, efforts were made to restrict recall to the last seven days while maintaining survey anonymity. It is crucial for future studies to encompass both the duration and quality of engaged behaviors, delving into aspects such as sleep quality and exercise intensity. Additionally, the assessment tools for CHI and other sociological factors posed limitations. While CHI assessment tools are widely adopted in China, their international comparability remains limited. The study also did not account for familial economic status and school environment, both of which could influence research outcomes. Hence, future research endeavors should encompass a combination of objective behavior assessment tools, such as accelerometers, alongside questionnaires, to better quantify and characterize children’s behaviors, CHI, and other pertinent familial characteristics like disposable income. Addressing these aspects comprehensively is imperative for future research endeavors.

### 4.5. Implications of the Research

From a theoretical perspective, this study is timely supplement to the literature related to the prevention and improvement of children’s hyperactive behavior and a further enrichment of the application of social ecology theory. First, this study actively applies social ecological theory to the improvement of children’s hyperactive behavior by applying the latest macro-environmental factor--COVID-19, in conjunction with the micro-environmental factor—parents—which directly affects children, to validate the effect of parental education level on children’s behavior. In addition, the change of certain behaviors of children will surely affect the change of another behavior, for most of the previous studies focus on individual behaviors (e.g., sleep), ignoring the holistic nature of children’s behaviors, while the present study breaks through the traditional method, considering children’s physical activity, sleep, and screen behaviors as the whole of the 24-h behaviors, and conducts multivariate model construction and research, which provides a more scientific and rigorous exploration of children’s healthy behaviors. It helps to sublimate the literature and theories related to children’s behavior optimization.

From a practical perspective, the study provides quantitative recommendations for optimizing children’s behavior that are tailored to parents with varying levels of educational attainment. The results of the study will not only alert parents and caregivers to the importance of daily child behavior management for the prevention and improvement of hyperactivity and aid parents’ understanding on how to improve their children’s health behaviors. Customized advice offers a convenient method for parents to directly intervene in their children’s lives, aligning with the notion that tailored interventions are pivotal for enhancing children’s daily health behaviors.

## 5. Conclusions

The 24-h behavior changes observed in children throughout the COVID-19 pandemic played a substantial role as a mediating factor in the relationship between Parental Education Attainment (PEA) and the Children’s Hyperactivity Index (CHI). The optimization of behaviors related to Children’s Physical Activity (CPA), Children’s Sleep (CSL), and Children’s Digital Media (CDM) demonstrated the potential to mitigate the adverse connections between PEA and CHI. In the case of parents with tertiary education (PTE), the inclination towards hyperactivity in children could be diminished by curbing excessive CDM and enhancing both CSL and safeguarded CPA. Similarly, parents without tertiary education (PWTE) can effectively decrease the risk of hyperactivity in their children by elevating CSL and CPA while concurrently reducing CDM. The study underscores the importance of parental vigilance and the deliberate optimization of children’s habits related to sleep, physical activity, and digital media, recognizing that the future health and overall well-being of the younger generation are of paramount importance for the nation.

## Figures and Tables

**Figure 1 healthcare-12-00516-f001:**
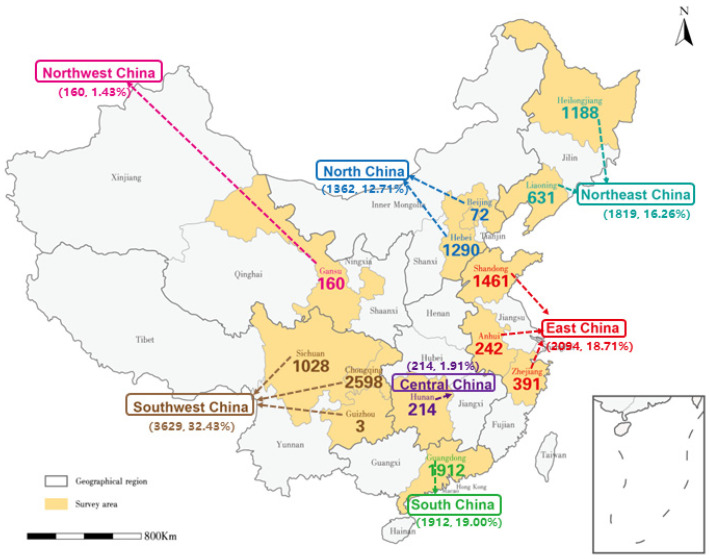
The numbers below the different cities represent the number of participants, and the numbers below the different geographic areas represent a combination of the total number and percentage of participants in the corresponding city.

**Figure 2 healthcare-12-00516-f002:**
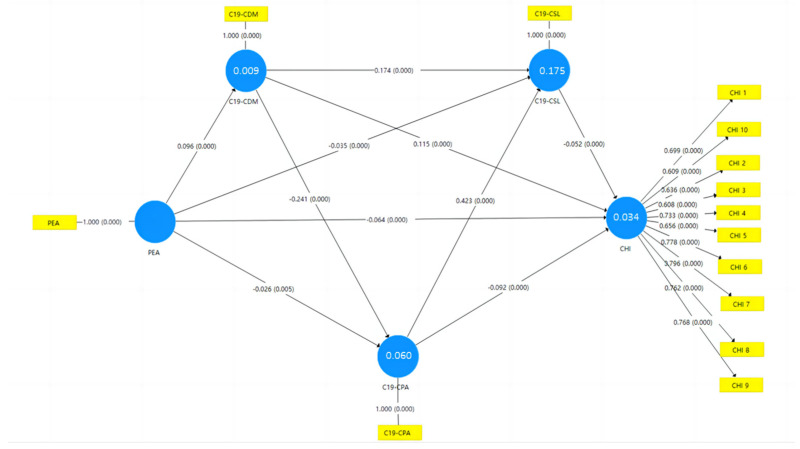
The numbers across the lines represent the coefficients and significance, and the numbers in the blue circle represent the R^2^ of the constructs; PEA = Parental Educational Attainment; C19 = a change because of the COVID-19 pandemic; CDM = Children’s Digital Media use; CSL = Children’s Sleep; CPA = Children’s Physical Activity; CHI = Children’s Hyperactivity Index.

**Figure 3 healthcare-12-00516-f003:**
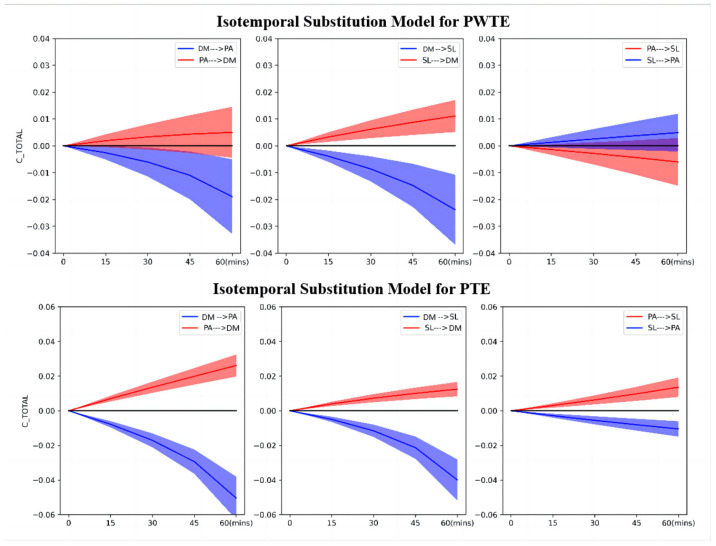
The graphs show the trends in CHI changes after the isotemporal substitution of children’s PA, SL, and DM for PWTE and PTE. Models were controlled for differences in age, gender, and geography. The shaded portions of the images depict the 95% confidence intervals for the respective child behaviors; DM = Digital Media use; PA = Physical Activity; SL = Sleep.

**Table 2 healthcare-12-00516-t002:** Results for measurement of the conceptual framework.

Constructs	Items	Factor Loadings	Std. D	T-Values	VIF	SCR	AVE	Alpha
CHI	CHI1	0.692	0.01	72.178	1.726	0.908	0.501	0.890
	CHI2	0.626	0.012	53.729	1.481			
	CHI3	0.601	0.012	49.923	1.621			
	CHI4	0.739	0.008	89.427	1.501			
	CHI5	0.651	0.011	61.413	1.947			
	CHI6	0.779	0.007	113.97	1.661			
	CHI7	0.806	0.007	123.849	2.082			
	CHI8	0.758	0.007	105.856	2.294			
	CHI9	0.775	0.007	108.436	1.999			
	CHI10	0.602	0.012	51.161	1.918			

CHI = Children’s Hyperactivity Index.

**Table 3 healthcare-12-00516-t003:** Discriminant validity of the Structural Equation Model.

**Fornell–Larcker Criterion**					
	**C19-CPA**	**C19-CSL**	**CHI_**	**PEA**	**C19-CDM**
C19-CPA	1.000				
C19-CSL	0.382	1.000			
CHI_	−0.138	−0.077	0.708		
PEA	−0.049	−0.039	−0.048	1.000	
C19-CDM	0.243	−0.068	−0.127	−0.096	1.000
**Heterotrait–Monotrait ratio of correlations (HTMT) criterion**					
	**C19-CPA**	**C19-CSL**	**CHI_**	**PEA**	**C19-CDM**
C19-CPA					
C19-CSL	0.382				
CHI_	0.135	0.077			
PEA	0.049	0.039	0.054		
C19-CDM	0.243	0.068	0.130	0.096	

C19 = a change because of COVID 19 disease; CPA = Children’s Physical Activity; CSL = Children’s Sleep; CHI = Children’s Hyperactivity Index; PEA = Parental Educational Attainment; CDM = Children’s Digital Media use.

**Table 4 healthcare-12-00516-t004:** Model fit Summary.

Statistical Tests	Estimation Model
SRMR	0.050
d_ULS	0.260
d_G	0.075
*X* ^2^	5101.864
NFI	0.899
RMS-theta	0.119

SRMR = Standardized Root Mean Square Residual; d_ULS = Unweighted Least Squares Discrepancy; d_G = Geodesic Discrepancy; NFI = Normed Fit Index; RMS = Root Mean Square.

**Table 5 healthcare-12-00516-t005:** Analysis of mediation effect path coefficient.

Effect Path		Beta	Std. D	T-Value	*p*-Values
Total effect	PEA -> CHI	−0.046	0.010	4.521	0.000 ***
Direct effect	PEA -> CHI	−0.064	0.010	6.330	0.000 ***
Total Indirect effect	PEA -> CHI	0.018	0.002	9.063	0.000 ***
Indirect effect	PEA -> C19-CSL -> CHI	0.002	0.001	3.046	0.002 **
	PEA -> C19-CPA -> CHI	0.002	0.001	2.600	0.009 **
	PEA -> C19-CDM -> CHI	0.011	0.001	7.708	0.000 ***
	PEA -> C19-CDM -> C19-CSL -> CHI	−0.001	0.000	4.141	0.000 ***
	PEA -> C19-CPA -> C19-CSL -> CHI	0.001	0.000	2.342	0.019 *
	PEA -> C19-CDM -> C19-CPA -> CHI	0.002	0.000	6.244	0.000 ***
	PEA -> C19-CDM -> C19-CPA -> C19-CSL -> CHI	0.001	0.000	4.192	0.000 ***

PEA = Parental Educational Attainment; CPA = Children’s Physical Activity; CSL = Children’s Sleep; CDM = Children’s Digital Media use; CHI = Children’s Hyperactivity Index; C19 = a change because of COVID-19 pandemic. *** means *p* < 0.001; ** means *p* < 0.01; * means *p* < 0.05.

**Table 6 healthcare-12-00516-t006:** Changes in CHI are predicted when reallocating time in PA, SB, and SL.

PEA	Path	Isotemporal Substitution Time			
		15 min	30 min	45 min	60 min
Parent Without Tertiary Education (BS = 0.485)	PA-DM	0.002 (0.000, 0.004)	0.003 (−0.001, 0.008)	0.004 (−0.003, 0.011)	0.005 (−0.004, 0.015)
	PA-SL	−0.001 (−0.003, 0.011)	−0.003 (−0.003, 0.011)	−0.004 (−0.003, 0.011)	−0.006 (−0.003, 0.011)
	SL-DM	0.003 (0.002, 0.005) *	0.006 (0.003, 0.010) *	0.009 (0.004, 0.013) *	0.011 (0.005, 0.017) *
	SL-PA	0.001 (−0.001, 0.003)	0.003 (−0.001, 0.006)	0.004 (−0.002, 0.009)	0.005 (−0.002, 0.012)
	DM-PA	−0.003 (−0.005, 0.000) *	−0.006 (−0.011, 0.001) *	−0.011 (−0.020, −0.002) *	−0.019 (−0.033, −0.005) *
	DM-SL	−0.004 (−0.006, −0.002) *	−0.009 (−0.013, −0.004) *	−0.015 (−0.023, −0.007) *	−0.024 (−0.037, −0.011) *
Parent with Tertiary Education (BS = 0.449)	PA-SL	0.003 (0.002, 0.004) *	0.006 (0.004, 0.009) *	0.010 (0.006, 0.014) *	0.014 (0.008, 0.019) *
	PA-DM	0.007 (0.005, 0.009) *	0.013 (0.010, 0.017) *	0.020 (0.015, 0.024) *	0.026 (0.020, 0.032) *
	SL-DM	0.004 (0.003, 0.005) *	0.007 (0.005, 0.009) *	0.010 (0.007, 0.013) *	0.012 (0.008, 0.016) *
	SL-PA	−0.003 (−0.004, −0.002) *	−0.006 (−0.008, −0.003) *	−0.008 (−0.012, −0.005) *	−0.010 (−0.015, −0.006) *
	DM-PA	−0.008 (−0.010, −0.006) *	−0.017 (−0.021, −0.013) *	−0.029 (−0.036, −0.022) *	−0.050 (−0.062, −0.038) *
	DM-SL	−0.005 (−0.006, −0.003) *	−0.011 (−0.015, −0.008) *	−0.021 (−0.027, −0.015) *	−0.040 (−0.051, −0.028) *

PA = Physical Activity; DM = Digital Media use; SL = Sleep; BS = Base Line. 95% confidence interval of change in the CHI is calculated. * means *p* < 0.05.

## Data Availability

The datasets produced and/or assessed during this study are not openly accessible to the public. Our commitment aligns with the research ethics approved by the institution, and we refrain from sharing the data with external parties. However, the Committee for Human Research Protection might consider granting access to the data for specific purposes, provided such requests are made in writing and accordance with the established procedures.

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
