# Peer review of "Associations between Parental Educational Attainment, Children’s 24-h Behaviors and Children’s Hyperactivity Behavior in the COVID-19 Pandemic"

_healthcare, 2024, doi:10.3390/healthcare12050516_

Round 1
Reviewer 1 Report
Comments and Suggestions for Authors
Well-prepared work.
The authors also quite precisely defined the shortcomings of the study. It could be valuable to add information to the analysis about which parent completed the questionnaire and whose de facto level of education is taken into account. Whether it was mother or father. Especially since, as research shows, fathers are usually very poorly exposed in such studies. Is it possible to verify the research group in this aspect?
Author Response
Thank you very much for taking the time to review this manuscript.
We agree with this comment. Therefore, we have supplemented the relevant data.In general, 2,211 fillers were fathers (19.7%), of which 584 were PWTE and 1,627 were PTE; 8,767 fillers were mothers (78.3%), of which 2,418 were PWTE and 6,049 were PTE; and 212 fillers were guardians (2%), of which 104 were PWTE and 108 were PTE. These additions are detailed in lines 260-262.
Reviewer 2 Report
Comments and Suggestions for Authors
Dear Authors
This study conducted to investigate the associations between parental educational attainment, children’s 24-hour behaviors and children’s hyperactivity behavior in the COVID-19 pandemic. This research topic is very interesting. However, it needs more revisions. Some suggestions are recommended for the authors’ consideration.
Minor issue
Abstract
Sort alphabetically in Keywords
Introduction
The authors should more explain the concept of parental educational attainment, children’s 24-hour behaviors and children’s hyperactivity behavior. Besides, the research gap in the article is not clear, and the authors should strengthen the gap illustration according to the prior research.
The literature review in this article seems brief. The authors need to supplement the literature and introduce the research status and main progress.
Line 3 and line 5: (1,2), (4,5) -> [1,2], [4,5], Healthcare journal guidelines of formatting must be followed in whole manuscript. Especially, (1) you have to revised the reference section based on Healthcare guidelines. (2) (refs) -> [refs]
Line 32: COVID-19 -> coronavirus disease 2019 (COVID-19)
Abbreviations should be defined in the first instance in whole manuscript.
The research flowchart in Fig. 1 is difficult to illustrate how could the study’s research design solve the research problem.
Methods
Between Line 128 and 132. You mentioned Appendix A in Line 128, and Appendix C in Line 132. However, I cannot see the Appendix B. Please insert it in main text.
Line 149: There should be a space before and after these mathematical symbols: ±, =, <, >, ≤, ≥, +, −, ÷, ×, ·, ≈, ∼, ∩, ∫, Π, Σ, and |.
Line 150: RMSEA, Abbreviations should be defined in the first instance in whole manuscript.
Please, you should insert foot-note of abbreviations (full name) in all supplementary tables and figure.
Discussion
The discussion lacks research findings: The authors should explain what new information was discovered from the measurement model and the structural equation model analysis.
I recommend that the authors make the requested modifications and resubmit the manuscript for review.
Comments on the Quality of English Language
I recommend that this manuscript should be edited by an English professional editor for more readable. There are several typo and grammatical errors.
Reviewer 3 Report
Comments and Suggestions for Authors
The topic of Associations between parental educational attainment, children's 24-hour behaviors, and children's hyperactivity behavior during the COVID-19 pandemic is significant. It highlights the correlation between parents' education, their awareness and skills in implementing activities that reduce hyperactivity in children (including physical activity), and a sedentary lifestyle. The presented research results confirm this relationship. However, the research methodology should be supplemented by specifying the selection of regions for the study (sample selection). Since this is related to the implementation of IISSAAR and CCASH programs, these programs should be discussed in more of detail.
There are also doubts about referring to IISAAR studies aimed at preschool children (line 114). It may be worthwhile to justify discussing the results of the study in the context of the school-age group (6-12 years – line 122) and briefly indicate the education system for children in China.
For an objective assessment of the research activities, statements such as "(...) and in China, it has demonstrated commendable reliability and validity" (lines 161-162) and "As vaccination rates increase and pandemic containment policies are gradually eased, macrosystems influencing children's health behaviors will inevitably evolve once again" (lines 383-384) should be removed.
Technically, the paper requires a few minor corrections; there is a period at the end of table and figure titles that should be removed. Additionally, the source of the study (e.g., own research) is missing.
Recommendation: The paper needs slight enhancements, particularly in the introduction, a more detailed discussion of IISSAAR and CCASH programs, and clarification of the method for selecting regions for the study.
Author Response
Thank you very much for taking the time to review this manuscript. Please find the detailed responses in the attachment.
